# Phylogenetic clustering networks among heterosexual migrants with new HIV diagnoses post-migration in Australia

Rachel Sacks-Davis[1,2]*, Doris Chibo[3], Elizabeth Peach[1], Eman Aleksic[1], Suzanne M. Crowe[1,4], Carol El Hayek[1,2], Tafireyi Marukutira[1,2], Nasra Higgins[5], Mark Stoove[1,2], Margaret Hellard[1,2,6,7]

1 Burnet Institute, Melbourne, Victoria, Australia, 2 Department of Epidemiology and Preventive Medicine, Monash University, Melbourne, Victoria, Australia, 3 Victorian Infectious Disease Reference Laboratory, Peter Doherty Institute, University of Melbourne, Melbourne, Victoria, Australia, 4 Department of Infectious Diseases, Monash University, Melbourne, Victoria, Australia, 5 Department of Health and Human Services, Melbourne, Victoria, Australia, 6 Hepatitis Services, Department of Infectious Diseases, The Alfred Hospital, Melbourne, Victoria, Australia, 7 Department of Medicine, University of Melbourne, Melbourne, Victoria, Australia

* rachel.sacks-davis@burnet.edu.au

**Data Availability Statement:** Anonymised HIV-1 sequence data has been submitted to Genbank with accession numbers MT748056-MT748757.

## Abstract

### Background

It is estimated that approximately half of new HIV diagnoses among heterosexual migrants in Victoria, Australia, were acquired post-migration. We investigated the characteristics of phylogenetic clusters in notified cases of HIV among heterosexual migrants.

### Methods

Partial HIV *pol* sequences obtained from routine clinical genotype tests were linked to Victorian HIV notifications with the following exposures listed on the notification form: heterosexual sexual contact, injecting drug use, bisexual sexual contact, male-to male sexual contact or heterosexual sexual contact in combination with injecting drug use, unknown exposure. Those with heterosexual sexual contact as the only exposure were the focus of this study, with the other exposures included to better understand transmission networks. Additional reference sequences were extracted from the Los Alamos database. Maximum likelihood methods were used to infer the phylogeny and the robustness of the resulting tree was assessed using bootstrap analysis. Phylogenetic clusters were defined on the basis of bootstrap and genetic distance.

### Results

HIV *pol* sequences were available for 332 of 445 HIV notifications attributed to only heterosexual sexual contact in Victoria from 2005–2014. Forty-three phylogenetic clusters containing at least one heterosexual migrant were detected, 30 (70%) of which were pairs. The characteristics of these phylogenetic clusters varied considerably by cluster size. Pairs were more likely to be composed of people living with HIV from a single country of birth (p = 0.032). Larger clusters (n≥3) were more likely to contain people born in Australian/New

Sufficient data to reproduce Tables 2 & 3 and Fig 3 have been included as a Supplementary file (S1 Dataset). Additional data can be requested from the Victorian Department of Health and Human Services.

**Funding:** The Burnet Institute gratefully acknowledges funding from the Victorian Operational Research Infrastructure fund. RSD, MS ad MH receive fellowship support from the Australian National Health and Medical Research Council (https://www.nhmrc.gov.au/). The funders had no role in study design, data collection and analysis, decision to publish, or preparation of the manuscript.

**Competing interests:** MH and MS has received investigator initiated funding from Gilead Sciences, AbbVie and Bristol Myers Squibb for research unrelated to this work. This does not alter our adherence to PLOS ONE policies on sharing data and materials.

Zealand (p = 0.002), migrants from more than one country of birth (p = 0.013) and viral sub-type-B, the most common subtype in Australia (p = 0.006). Pairs were significantly more likely to contain females (p = 0.037) and less likely to include HIV diagnoses with male-to-male sexual contact reported as a possible exposure (p<0.001) compared to larger clusters (n≥3).

## Conclusion

Migrants appear to be at elevated risk of HIV acquisition, in part due to intimate relationships between migrants from the same country of origin, and in part due to risks associated with the broader Australian HIV epidemic. However, there was no evidence of large transmission clusters driven by heterosexual transmission between migrants. A multipronged approach to prevention of HIV among migrants is warranted.

## Introduction

Heterosexual sexual transmission of HIV accounts for approximately 20% of new HIV notifi-cations in Australia annually and has remained stable from 2008–2017, with migrants making up approximately 40% of these notifications [1–6]. Although many HIV infections among those migrating from high prevalence countries to low prevalence countries are likely to be acquired before migration, migrants may also be vulnerable to acquiring HIV in the destina-tion country [7, 8]. Results from a study of HIV in heterosexual migrants in the United King-dom (UK) from 2004–2010 using CD4 cell counts at diagnosis and a subsequent study migrants diagnosed at 57 clinics across Europe between 2013–2015 using a combination of clinical and self-report data, indicated that 33% and 63% of diagnosed HIV cases, respectively, were likely to have been acquired post-migration [9, 10]. We previously applied a CD4 count model to data from HIV notifications with heterosexual exposure to HIV in the Australian state of Victoria, with the results suggesting that approximately 50% of migrants were likely to have acquired their HIV post-migration to Australia [3].

Molecular epidemiology, including phylogenetic analysis, is increasingly being used as to monitor and characterise HIV transmission and inform public health and prevention responses [11–14]. Molecular epidemiology has also been used to demonstrate increases in non-B subtypes in Australia and several European countries [15–23], and to estimate the num-ber of infections likely to have been acquired post-migration in some cases [22]. However, these studies have not examined the characteristics of phylogenetic clusters, beyond using them to determine acquisition post-migration. A mathematical model of HIV infection in the Netherlands found that highly assortative sexual mixing between migrants who came from the same region of origin resulted in higher HIV prevalence among migrants [24]. We used phylo-genetic analysis and subtyping of routine HIV *pol* sequencing (undertaken as part of clinical care) to better understand heterosexual HIV risk and transmission among migrants and explore the role of networks in transmission.

## Methods

### Study population

The key population of interest was migrants in the Australian state of Victoria who were newly diagnosed with HIV and reported heterosexual sexual contact as the route of transmission.

Victoria is Australia's second most populous state and has the second highest number of people diagnosed with HIV and new HIV diagnoses [25, 26].

We included all migrants with new HIV diagnoses aged ≥18 years notified to the Victorian Department of Health and Human Services, where the likely exposure to HIV was recorded as heterosexual sexual contact. People born in any country other than Australia or New Zealand, were classified as migrants, irrespective of citizenship status, and Australian and New Zealand-born individuals were classified as non-migrants. Consistent with previous analyses, New Zealanders were grouped with Australian-born people [3]. In addition, the following notifications were used to facilitate characterisation of phylogenetic clusters among cases classified as heterosexually acquired: all notifications in non-migrants aged≥18 years attributed to heterosexual sexual contact and all notifications attributed to bisexual sexual contact or injecting drug use (including those attributed to injecting drug use and male-to-male sex [MSM]) in those ≥18 years irrespective of migrant status.

## Data sources

The Victorian Department of Health and Human Services receives notifications of HIV from both the laboratories performing the test and the diagnosing doctors, as mandated by the Public Health and Wellbeing Act 2008 and its associated regulations. Notification data were obtained from 1st January 2000 (as very little sequencing was done prior to 2000) to 30th of June 2014. From 2000–2004 relatively few HIV notifications were sequenced. Therefore, although these sequences were included in the phylogenetic analysis to allow for possible clustering with later notifications, notifications from 2005 onward were the focus of the analysis.

## Patient characteristics

Patient and associated diagnosis characteristics including date of diagnosis, patient demographics (sex and date and country of birth), clinical characteristics at the time of diagnosis (CD4 count, any reported symptomatology), HIV testing history (date and result of the previous HIV test), possible route/s of exposure, likely country of exposure and year of arrival to Australia (for people born outside of Australia or New Zealand) were extracted from notifications data. Multiple HIV exposures can be selected on the notification form: for the purpose of these analyses, exposures were categorised as heterosexual sex (where heterosexual sexual contact was the only reported exposure), bisexual (where MSM and heterosexual sexual contact were both listed on the form), heterosexual sex with injecting drug use, MSM with injecting drug use and other/unknown.

## HIV sequence data

Sanger sequencing of the HIV *pol* gene is used in routine clinical practice to determine antiretroviral drug susceptibility and HIV subtype. Pre-treatment HIV *pol* sequences were identified for the period 2000–2014 in the Victorian Infectious Diseases Reference Laboratory (VIDRL) database and the Burnet Institute Clinical Research Laboratory HIV database, where clinical samples from patients with HIV infection receiving their clinical care at The Alfred hospital (a Victorian HIV care service), were routinely sequenced as part of HIV genotype testing during this period. Both laboratories participated in quality assurance programs. Data were linked with HIV notifications by matching name codes (the first two letters of cases' first name and surname) and date of birth, recorded in all three datasets.

Reference sequences were selected from the Los Alamos database using a stratified random sample, with strata defined by country of birth and viral subtype. Where available, three

sequences were selected from each country of birth and viral subtype combination represented in the study sample. The final sample included 177 reference sequences.

## Classifying notifications on the basis of the probable timing of HIV acquisition relative to migration

For our analyses, evidence of probable place of HIV acquisition was classified as "strong", "medium" or "weak" on the basis of testing history, laboratory evidence of recent infection and CD4 count at diagnosis. Those individuals with a clinic and/or laboratory-confirmed previous negative HIV test post-migration to Australia and those with laboratory evidence of recent infection (group IV indeterminate Western blot and HIV detected by virological assay) post-migration to Australia were classified as having strong evidence of HIV acquisition post-migration. Those with a self-reported previous negative HIV test post-migration to Australia were classified as having medium evidence of HIV acquisition post-migration.

Weak evidence of place of acquisition was classified on the basis of their CD4 count at diagnosis. We adopted the formulae used in the UK's HIV & AIDS New Diagnoses & Deaths Database [3, 9] which estimates time since HIV acquisition based on modelled estimates of the median estimated CD4 counts at HIV acquisition and the median rate of CD4 decline after diagnosis. We used known CD4 count at diagnosis from our dataset to estimate time between HIV exposure and diagnosis which we subtracted from date of diagnosis to estimate date of acquisition and upper and lower confidence intervals for this estimate [3, 9]. If the year of arrival to Australia was before the lower bound confidence interval of the estimated year of acquisition, a HIV case was classified as weak evidence of acquisition post-migration. All other cases including those where the bounds of the confidence interval around estimated year of acquisition included year of arrival were classified as weak evidence of having acquired their infection before migration [3].

Those notifications in migrants that could not be classified using any of the methods above (no evidence of recent infection, no negative test in Australia and either CD4 count or year of migration not available), location of HIV acquisition were classified as unknown.

## Sequencing methods

A 1035 base-pair product spanning the entire coding region for protease (PR) and the first 246 codons of reverse transcriptase (RT) was amplified from HIV-1-specific RNA derived from 500 μL of plasma and sequenced using the ABI Prism reagents, hardware and software (Applied Biosystems, Foster, CA; HXB2 co-ordinates of the sequence dataset set are 2253–3287). The methods have been described in detail previously [15]. HIV strains were initially subtyped based on their *pol* sequences using the Stanford HIV database (http://hivdb.stanford.edu/). Subtype assignment was confirmed by submitting sequences to the Los Alamos database (http://www.hiv.lanl.gov) and the NCBI HIV genotyping tool (http://www.ncbi.nlm.nih.gov/projects/genotyping/). If subtype assignment was unclear due to potential recombination, Recombinant Identification Program (https://www.hiv.lanl.gov/content/sequence/RIP/RIP.html) and Jumping Profile Hidden Markov Model (http://jphmm.gobics.de/) were used to assign the subtype.

## Phylogenetic analysis

Sequences were aligned using the Bio Edit tool (mbio.ncsu.edu/BioEdit). Codons associated with drug resistance were removed to avoid clustering due to convergent evolution (hivdb.stanford.edu/s/who). The nucleotide substitution model was selected based on the Akaike Information Criteria. Maximum likelihood methods used to construct a phylogenetic tree

using Mega 6.0 [27] based on the general time reversible nucleotide substitution model with gamma distribution and a proportion of invariable sites. The robustness of the resulting tree was assessed using bootstrap with 1000 replicates.

Phylogenetic clusters were defined on the basis of genetic distance (<0.045) and bootstrap support (>0.95) using the command line version of ClusterPicker version 1.2. Genetic distance was defined using ClusterPicker's "ambiguity" method which is the p-distance for A, C, T, G sites and sites with IUPAC ambiguity codes. The characteristics of phylogenetic clusters were visualised using Pajek 5.07.

## Statistical analysis

Chi-squared (categorical variables) and Kruskal-Wallis tests (numeric variables) were used to assess differences in notification characteristics between migrants and non-migrants. Potential associations between phylogenetic cluster size and the composition of the clusters were assessed using Fisher's exact (categorical variables) and Kruskal-Wallis tests (numeric variables). The probability that phylogenetic clusters containing migrants from the same country of origin would be observed by chance was assessed by calculating the binomial probability of same country clusters being observed given that the largest group of migrants migrating from the same country was 45 and 876 sequences were included in the phylogenetic analysis.

We assessed factors associated with phylogenetic clustering using univariable and multivariable logistic regression associated with phylogenetic clustering, with a backward selection approach used to build the multivariable model which initially included variables whose p-value was <0.25 in univariable regression. The cut-off for statistical significance for all analyses was p<0.05.

## Ethics

Approval for the study including a waiver of informed consent for use of retrospective data was granted by The Alfred Office of Ethics and Research Governance (project 218/14).

## Results

There were 445 new HIV diagnoses in Victoria between 2005–2014 in people reporting heterosexual sex as the only exposure (Fig 1). Of these, 332 (75%) had *pol* sequence data available. The characteristics of new HIV diagnoses with sequence data available are described in Table 1. Compared to individuals born in Australia/New Zealand, migrants were on average younger at notification (median 36 years vs 42 years, p = 0.013) and less likely to have evidence of newly acquired infection (9% vs 21%, p = 0.003). Among migrants, the most common region of birth was sub-Saharan Africa (35%) followed by South-East Asia (22%). The HIV-1 subtype distribution was substantially different among migrants (22% B, 41% C, 24% CRF01_AE, 13% other) compared to Australian/New Zealand-born individuals (63% B, 12% C, 21% CRF01_AE, 4% other; p<0.001, Table 1). Availability of HIV partial *pol* sequence did not differ by age, sex, country of birth (Australian/New Zealand born vs migrant), or CD4 count.

### Phylogenetic clustering

Overall, 876 sequences were included in the phylogenetic analysis, including reference sequences (Figs 1 and 2). Of these, 258 (29%) were in 92 phylogenetic clusters. Of the 332 notifications from 2005 onward where heterosexual sex was listed as the only exposure and *pol* sequences were available, 206 (62%) were from migrants and 126 (38%) were Australian/NZ born, with 118 (36%) samples in 65 phylogenetic clusters. Of these 118 samples identified in a

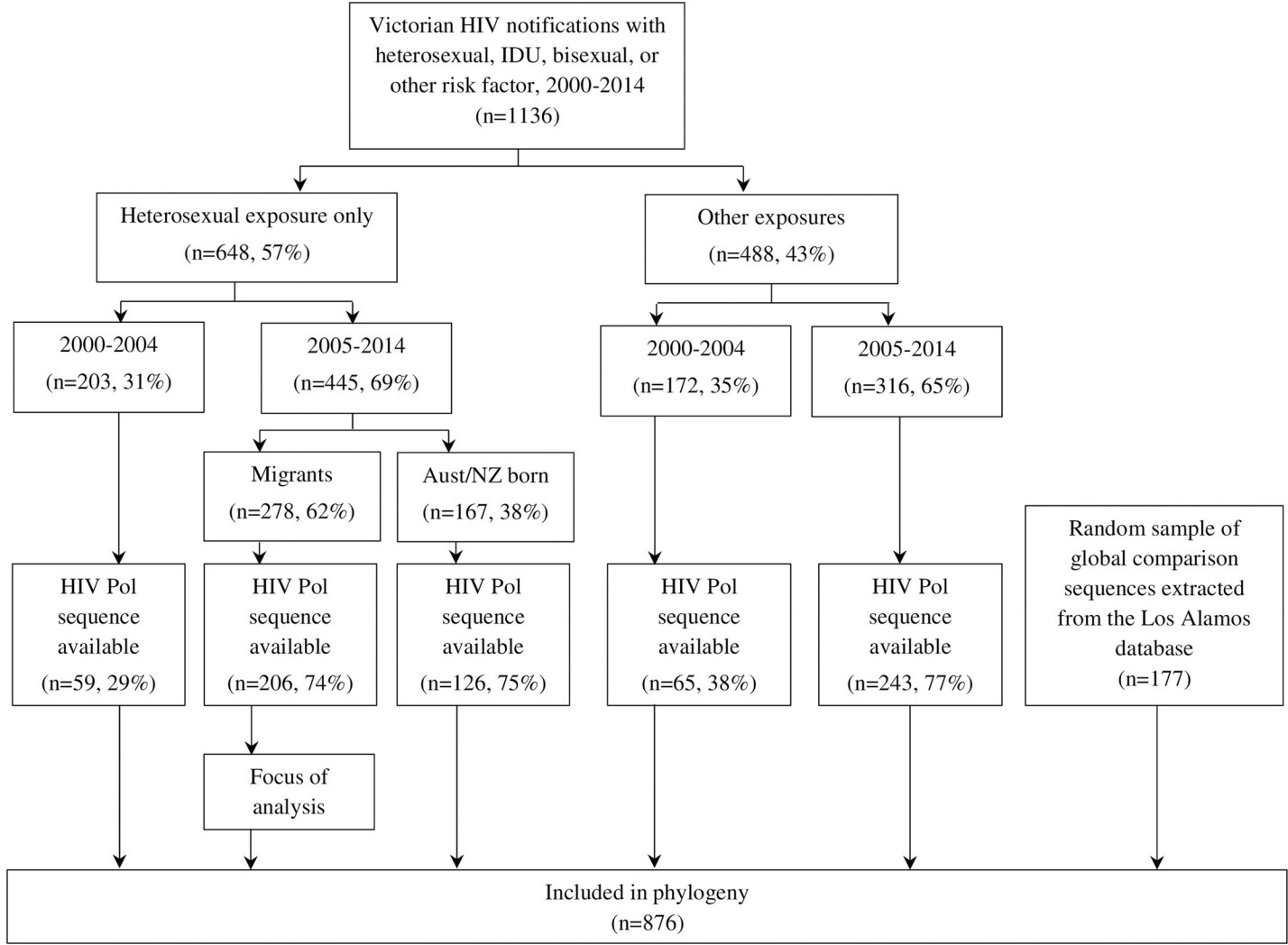

**Fig 1. Flow chart of study inclusion.**

cluster, 66 (56%) were migrants and 52 (44%) were Australian/NZ-born notifications. Based on multivariable analysis, newly acquired infection, region of birth ("Other" compared to sub-Saharan Africa) and B- subtype were independent predictors of phylogenetic clustering (S1 Table).

Australian HIV notifications include 206 in migrants to Australia with heterosexual exposure from 2005–2014 and 493 other Australian HIV notifications from 2000–1014. International reference sequences (n = 177) were sampled from the Los Alamos database. Phylogenetic clusters, identified on the basis of bootstrap value and genetic distance, are coloured red.

Of the 65 phylogenetic clusters identified, 43 contained at least one migrant who reported heterosexual sex as the only mode of transmission (Fig 3). The size of the clusters ranged from 2–10 with 30 clusters (70%) consisting of pairs (cluster size = 2) (Fig 3). The characteristics of the pairs were considerably different to those of the larger clusters (Table 2, Fig 3). Compared to clusters larger than two, pairs were more likely to include at least one female (87% vs. 54% of clusters larger than 2, p = 0.037) and more likely to be composed of migrants from the same

**Table 1. Characteristics of 332 HIV notifications with heterosexual exposure category and *pol* sequences available, 2005–2014.**

| | Migrants | Australian/NZ-born | p-value[a] |
|---|---|---|---|
| **N** | 206 | 126 | |
| **Sex at birth** | | | |
| Male | 109 (53) | 79 (63) | 0.081 |
| Female | 97 (47) | 47 (37) | |
| **Median age at HIV diagnosis (IQR)** | 36.1 (30.1–46.8) | 42.3 (31.5–50.6) | 0.013 |
| **Year of HIV diagnosis** | | | |
| 2005–2009 | 108 (52) | 61 (48) | 0.478 |
| 2010–2014 | 98 (48) | 65 (52) | |
| **Median CD4 count at HIV diagnosis (IQR)[b]** | 243 (70–430) | 304 (102–538) | 0.068 |
| **Evidence of newly acquired infection** | | | |
| Yes | 19 (9) | 26 (21) | 0.003 |
| No | 187 (91) | 100 (79) | |
| **Region of birth** | | | N/A |
| North Africa and Middle East | 13 (6) | - | |
| Americas | 5 (2) | - | |
| North East Asia | 7 (3) | - | |
| South East Asia | 46 (22) | - | |
| Southern and Central Asia | 17 (8) | - | |
| North and West Europe | 20 (10) | - | |
| South and East Europe | 11 (5) | - | |
| Oceania | 7 (3) | 126 (100) | |
| Sub-Saharan Africa | 71 (35) | - | |
| Unknown | 9 (4) | - | |
| **HIV-1 subtype** | | | |
| B | 45 (22) | 80 (63) | <0.001 |
| C | 85 (41) | 15 (12) | |
| CRF01_AE | 49 (24) | 26 (21) | |
| Other | 27 (13) | 5 (4) | |

[a]*p-value* for difference between migrants and Australian born cases.
[b]35 migrants and 32 Australian-born cases missing CD4 count data.

country of origin (40% vs 0% of clusters larger than 2; p = 0.032). Larger clusters were more likely to include bisexual/MSM-IDU (70% vs. 17% of pairs; p<0.001), people born in Australia/New Zealand (including people from multiple countries of origin) (92% vs 40% of pairs; p = 0.002) and people from multiple countries of origin that do not include Australia (62% vs. 21% of pairs, p = 0.013) The proportion of clusters that included heterosexual migrants with any (p = 0.890) or strong (p = 1.000) evidence of HIV acquisition after migration did not differ by cluster size (Table 2).

Of the 43 phylogenetic clusters including 67 migrants who reported heterosexual sex as their only mode of transmission, 22 individuals (33%) were in 12 clusters with only migrants from the same country (Tables 2 & 3). Of the 22, HIV acquisition was estimated to have occurred before migration for nine, seven after migration and six were unknown (Table 3). Given that there were 876 sequences in the analysis and the largest group migrating from the same country was 45, the probability of observing a same-country pair by chance was p = 0.003. Therefore, the observation of 12 same country pairs overall and seven same country pairs with evidence of infection acquired after migration, were statistically significant (both

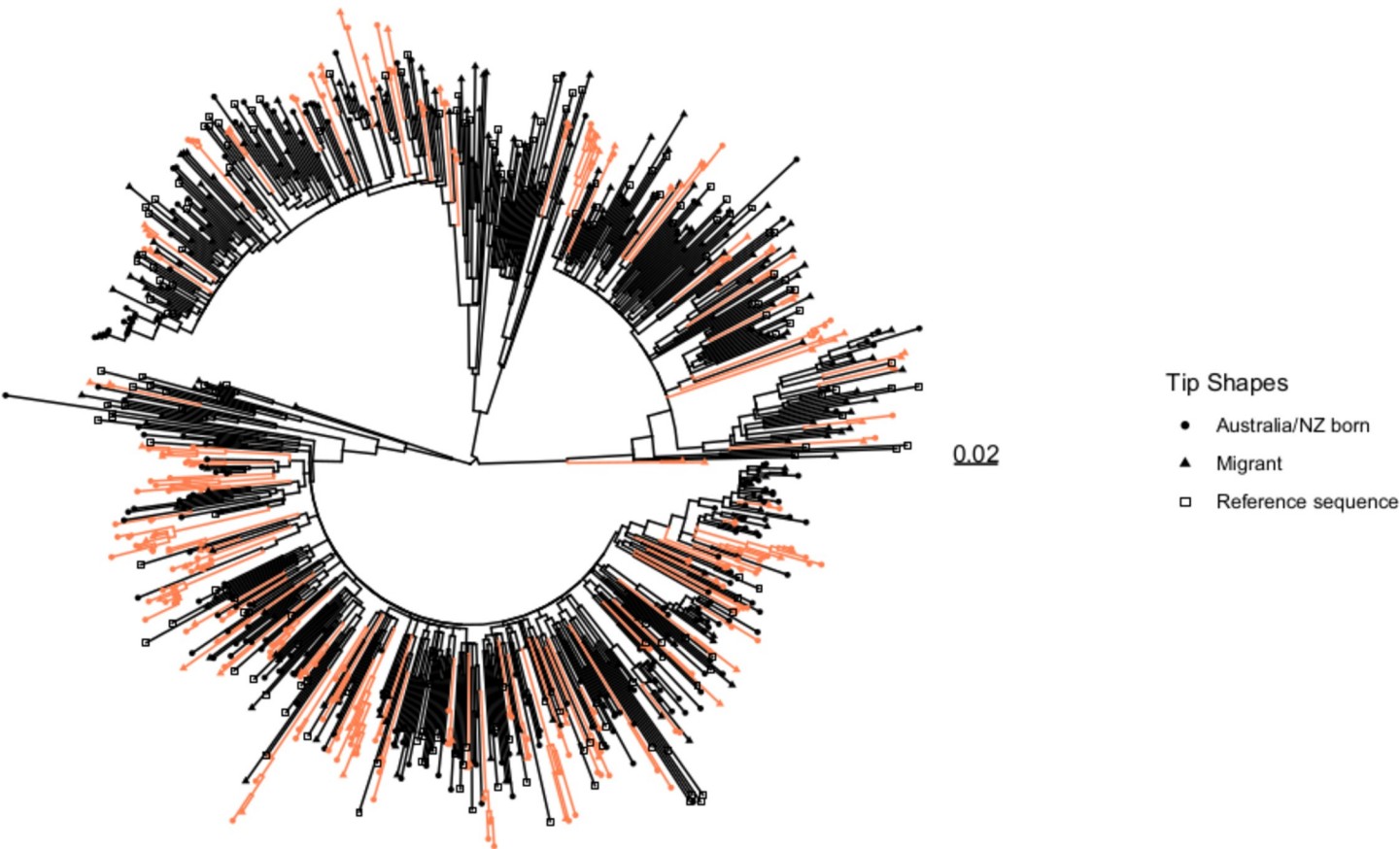

**Fig 2. Maximum likelihood phylogeny of HIV *pol* sequences from Australian HIV notifications and international reference sequences.**

p<0.001). Nonetheless, these observed infections in single-country clusters accounted for a minority (7/29; 24%) of all Australian acquired infections among migrants.

## Discussion

This study examined the characteristics of phylogenetic clusters involving heterosexual migrants living in Victoria, Australia from 2005–2014. The majority (70%) of 43 clusters that included at least one migrant with heterosexual risk were pairs and all clusters with more than one migrant from a single country of origin were pairs. The number of same country migrant pairs that we observed was significantly greater than expected by chance, including among pairs with evidence of post-migration acquisition of HIV. Whilst this suggests that the sexual networks among migrants from the same country of origin does contribute to the risk of HIV infection after migration, the overall number of same country-of-origin clusters was small and confined to pairs rather than larger clusters. In addition, the risk of heterosexually acquired infection among migrants appears to be partially attributable to sex between migrants from the same country (within couples) and partially attributable to risk from sex with those born in Australian/New Zealand, including those reporting male-to-male sex.

The eight clusters consisting of four or more notifications included mainly men, people born in Australian/NZ and migrants from multiple countries of origin, and those reporting male-to-male sex as a possible route of HIV acquisition. These clusters were also mainly viral subtype B, the predominant HIV subtype transmitted through male-to-male sex in Australia

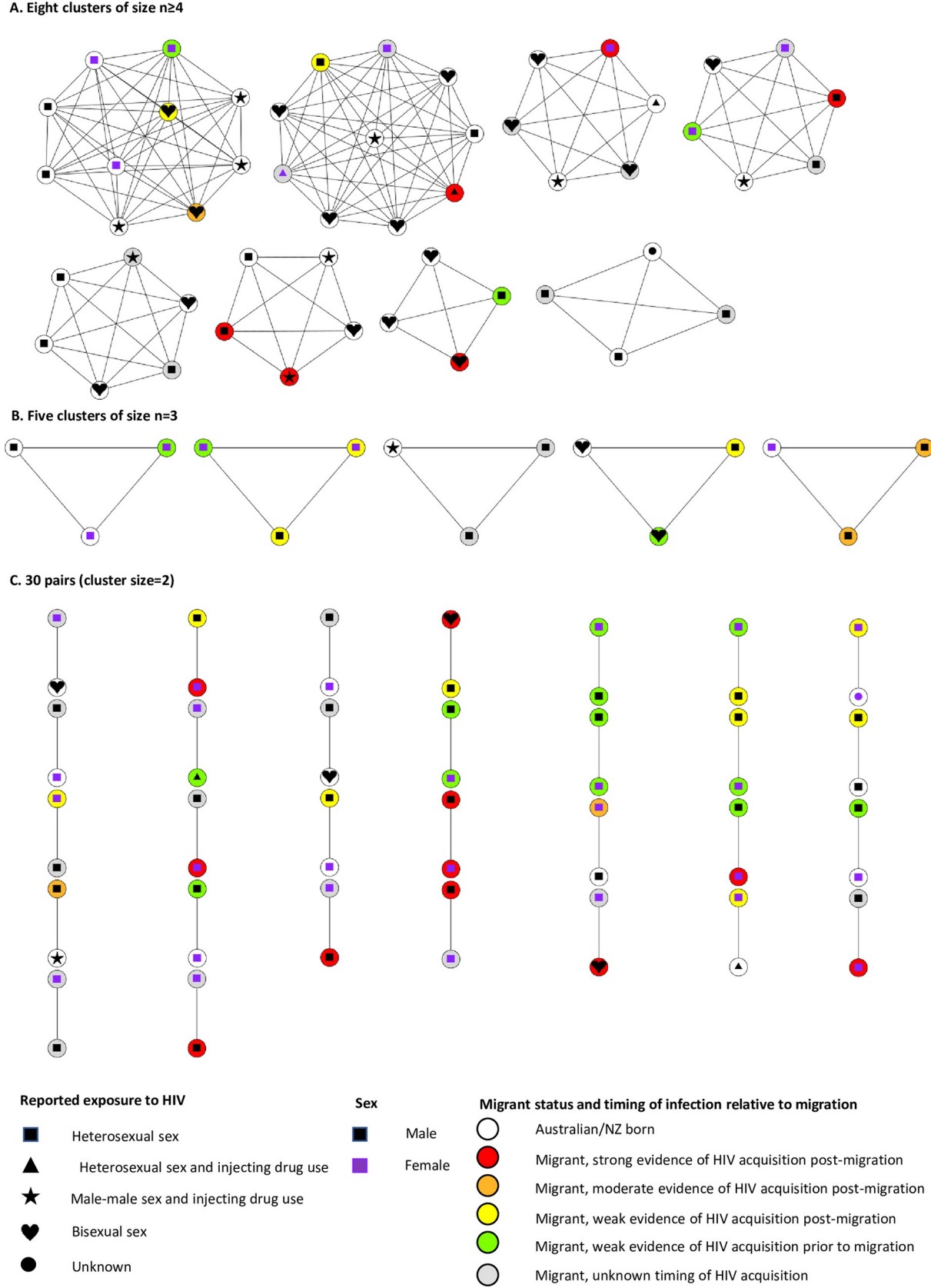

A. Eight clusters of size n≥4

B. Five clusters of size n=3

C. 30 pairs (cluster size=2)

**Reported exposure to HIV**

■ Heterosexual sex

▲ Heterosexual sex and injecting drug use

★ Male-male sex and injecting drug use

♥ Bisexual sex

● Unknown

**Sex**

■ Male

■ Female

**Migrant status and timing of infection relative to migration**

○ Australian/NZ born

● Migrant, strong evidence of HIV acquisition post-migration

● Migrant, moderate evidence of HIV acquisition post-migration

● Migrant, weak evidence of HIV acquisition post-migration

● Migrant, weak evidence of HIV acquisition prior to migration

● Migrant, unknown timing of HIV acquisition

**Fig 3. The characteristics of 43 phylogenetic clusters containing at least one migrant with heterosexual exposure to HIV.** Each circle represents an HIV notification. Lines connect notifications within the same cluster. The symbols in the centre of each circle represent the exposure to HIV listed on the notification form. The colour of the symbol in the centre of each circle represents sex. The fill colour of the circle represents migrant status, estimate and level of evidence for timing of HIV acquisition relative to migration.

[5]. These findings suggest that migrants in larger clusters are predominantly acquiring their HIV after migration and HIV risk is occurring through sexual contact with people born in Australia/New Zealand and may include male-to-male sex. Further work is required to see how well-connected sequences from migrants are with those from the Australian gay community. It is possible that some of the reported heterosexual exposures in these clusters were due to male-to-male sexual exposure that was not reported due to social desirability bias, which

**Table 2. Characteristics of 43 phylogenetic clusters containing at least one heterosexual migrant with new HIV diagnosis, 2005–2014.**

|  | Size of cluster | | | p-value |
|---|---|---|---|---|
|  | **2** | **3** | **4–10** |  |
| **Total number of clusters** | 30 | 5 | 8 |  |
| ***Sex of cluster members*** |  |  |  |  |
| Cluster includes females: n (%) | 26 (87) | 3 (60) | 4 (50) | 0.037 |
| Proportion of cluster female: median (IQR) | 0.5 (0.5–0.5) | 0.3 (0.0–0.7) | 0.1 (0.0–0.3) | 0.001 |
| ***Sexual orientation and sexual behaviour of cluster members*** |  |  |  |  |
| Cluster includes MSM: n (%) | 5 (17) | 2 (40) | 7 (88) | <0.001 |
| Cluster includes bisexuals: n (%) | 4 (13) | 1 (20) | 7 (88) | <0.001 |
| Proportion of cluster MSM: median (IQR) | 0.0 (0.0–0.0) | 0.0 (0.0–0.3) | 0.5 (0.4–0.6) | <0.001 |
| ***Injecting drug use in the cluster*** |  |  |  |  |
| Cluster includes PWID (including MSM who inject drugs): n (%) | 2 (7) | 1 (20) | 7 (88) | <0.001 |
| Cluster includes PWID (excluding MSM): n (%) | 1 (3) | 0 (0) | 1 (13) | 0.518 |
| Proportion of cluster reporting injecting drug use (includes MSM): median (IQR) | 0.0 (0.0–0.0) | 0.0 (0.0–0.0) | 0.2 (0.1–0.3) | <0.001 |
| ***Country of origin in the cluster*** |  |  |  |  |
| Cluster composed only of migrants from one country of origin: n (%) | 12 (40) | 0 (0) | 0 (0) | 0.032 |
| Cluster includes Australian/NZ born: n (%) | 12 (40) | 4 (80) | 8 (100) | 0.002 |
| Cluster includes migrants from two or more countries of origin: n (%) | 6 (21) | 2 (40) | 6 (75) | 0.013 |
| Number of different countries of origin per cluster (includes Aust/NZ): median (IQR) | 2 (1–2) | 2 (2–2) | 3 (3–4) | <0.001 |
| Proportion of cluster Australian/NZ born: median (IQR) | 0.0 (0.0–0.5) | 0.3 (0.3–0.3) | 0.6 (0.5–0.6) | 0.002 |
| ***Viral subtype of cluster*** |  |  |  |  |
| B: n (%) | 8 (27) | 2 (40) | 7 (88) | 0.006 |
| ***Timing of HIV infection among heterosexual migrants in the cluster*** |  |  |  |  |
| Cluster includes heterosexual migrants with any evidence of HIV infection after migration: n (%) | 18 (60) | 3 (60) | 4 (50) | 0.890 |
| Cluster includes heterosexual migrants with strong evidence of HIV infection after migration: n (%) | 10 (33) | 1 (20) | 3 (38) | 1.000 |

The denominators for all percentages are the total number of clusters in the column. p-values for differences between clusters of different sizes (n = 2, n = 3, n≥4) based on Fisher's exact test for categorical variables and Kruskal Wallis test for continuous variables.

**Table 3. Single country of origin clustering among 67 heterosexual migrants with new HIV notification 2005–2014 who are members of phylogenetic clusters, by timing of HIV acquisition relative to migration.**

|  | Number of notifications | Cluster with only same country migrants n (%) |
|---|---|---|
| Total | 67 | 22 (33) |
| *Strong evidence acquired after migration* | 12 | 3 (25) |
| *Moderate evidence acquired after migration* | 4 | 0 (0) |
| *Weak evidence acquired after migration* | 13 | 4 (31) |
| *Weak evidence acquired before migration* | 16 | 9 (56) |
| *Unknown* | 22 | 6 (27) |

may be higher among migrants from countries or cultures where male-to-male sex is stigmatised. A recent study of Australian HIV diagnoses in migrants found that diagnoses attributable to male-to-male sex appeared to be increasing in migrants from several regions of origin, although this was based only on a comparison between two periods rather than an analysis of trends [6].

To the best of our knowledge, the only published study of HIV transmission networks in migrants is a mathematical modelling study of the heterosexual HIV epidemic in the Netherlands, which investigated the effects of sexual mixing patterns on the epidemic [24]. Modelled estimates based on empirical data on sexual mixing and sexual behaviour from cross-sectional surveys suggested that the majority of HIV infections in migrants from high prevalence countries were acquired in the Netherlands, but the epidemic was very stable due to low levels of risk behaviour among migrants and highly assortative mixing within migrants from the same region. That study focussed on only migrants from high and medium HIV prevalence regions whereas our study was on all heterosexual migrant notifications. Sexual mixing and sexual behaviour among migrants to Australia may differ from migrants to the Netherlands. Nonetheless, a key finding of the modelling study was that the heterosexual epidemic in the Netherlands was stable due to low levels of risk behaviour among migrants from high prevalence countries. This finding is consistent with the findings of our study in that phylogenetic clusters consistent with heterosexual transmission alone were small (all n≤3 and most n = 2).

The results of our study suggest the risk of HIV acquisition among heterosexual migrants to Australia post-migration is from a variety of sources; for some this is within country of origin networks including transmission before or after migration and for others it is following migration and is related to the Australian predominantly MSM HIV epidemic. The proportion with evidence of acquisition after migration did not vary with cluster size, highlighting that HIV risk for migrants after migration is hybrid, both from small clusters which are likely to represent couples that are often composed of migrants from the same country of birth, and from risks associated with the Australian epidemic, which is largely driven by male-to-male sex. Within country of origin transmission networks accounted for a minority of transmission and the infections that occurred post-migration may have been preventable by timely diagnosis and ART treatment of partners living with HIV. Unfortunately previous findings from Australia and other high income countries are that migrants are at increased risk of delayed diagnosis [3, 28–32] and delayed ART treatment [33]. While there remains structural barriers to HIV testing and care for migrants in Australia, including a lack of access for some migrants to universal health coverage [33], qualitative research has also found that diverse groups of migrants to Australia perceive HIV risk in Australia to be low [34]. In addition to changes to eligibility universal health coverage that include migrants, non-stigmatising and culturally appropriate education about the risk of HIV and STIs in Australia in migrant groups may also be needed.

## Limitations

Analysis of partial *pol* sequences obtained through routine clinical practice continues to be a key strategy for population genomic studies in HIV [11]. However, while phylogenetic clustering implies closely related infection, it does not necessarily imply direct transmission. It is possible that there were other infections in the transmission networks that have not yet been diagnosed, were diagnosed in another state or country and therefore were not notified in Victoria, were diagnosed prior to the study period, or did not have a *pol* sequence available. It is possible that some cases that were classified as having acquired HIV infection post-migration to Australia were exposed to HIV whilst travelling back to their country/region of birth or other regions abroad, which was not measured in this study. Furthermore, it was not possible to describe the possible transmission networks of those who were not in phylogenetic clusters.

## Conclusions

To the best of our knowledge, this is the first study to investigate the characteristics of possible HIV transmission networks specifically among heterosexual migrants. We found that post-migration risk of HIV infection in heterosexual migrants was attributable in part to sexual networks among migrants from the same country of origin (mostly pairs), but also attributable to larger sexual networks that included those born in Australia and New Zealand and those reporting male-to-male sex. A multipronged approach to prevention is warranted including promoting timely diagnosis and treatment of HIV infected migrants and non-stigmatising culturally appropriate education about HIV risk in Australia for migrants.

## Supporting information

**S1 Table. Unadjusted and adjusted odds of phylogenetic clustering among partial *pol* sequences in 332 HIV 1 notifications 2005–2014 in heterosexuals with *pol* sequences available.**
(DOCX)

**S1 Dataset. Minimum data required to reproduce Tables 2 & 3 and Fig 3.**
(XLS)

## Acknowledgments

We would like to thank Matthew Kaye for his assistance identifying HIV sequence data in the VIDRL database, Imogen Elsum for her assistance identifying and collating HIV sequence data in the Burnet HIV sequence database and Clarissa Moreira for her assistance extracting HIV notification characteristics data from the HIV surveillance database.

## Author Contributions

**Conceptualization:** Rachel Sacks-Davis, Elizabeth Peach, Margaret Hellard.

**Data curation:** Doris Chibo, Elizabeth Peach, Eman Aleksic, Carol El Hayek, Tafireyi Marukutira.

**Formal analysis:** Rachel Sacks-Davis, Doris Chibo, Elizabeth Peach.

**Investigation:** Doris Chibo, Eman Aleksic, Suzanne M. Crowe.

**Methodology:** Rachel Sacks-Davis, Doris Chibo, Elizabeth Peach, Suzanne M. Crowe, Carol El Hayek, Nasra Higgins, Mark Stoove, Margaret Hellard.

**Project administration:** Carol El Hayek, Nasra Higgins.

**Supervision:** Mark Stoove, Margaret Hellard.

**Visualization:** Rachel Sacks-Davis.

**Writing – original draft:** Rachel Sacks-Davis.

**Writing – review & editing:** Doris Chibo, Eman Aleksic, Suzanne M. Crowe, Carol El Hayek, Tafireyi Marukutira, Nasra Higgins, Mark Stoove, Margaret Hellard.

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
