## [Decision Letter · Decision Letter 0]

29 May 2020

PONE-D-20-12180

Phylogenetic clustering networks among heterosexual migrants with new HIV diagnoses post-migration in Australia

PLOS ONE

Dear Dr. Sacks-Davis,

Thank you for submitting your manuscript to PLOS ONE. After careful consideration, we feel that it has merit but does not fully meet PLOS ONE’s publication criteria as it currently stands. Therefore, we invite you to submit a revised version of the manuscript that addresses the points raised during the review process.

We look forward to receiving your revised manuscript.

Kind regards,

Jason Blackard, PhD

Academic Editor

PLOS ONE

Additional Editor Comments (if provided):

This is a study of HIV transmission networks in Australia.

The overall study rationale and methods are well described.  

It is a bit surprising that phylogenies were not investigated with a more robust approach such as Bayesian inference.

The manuscript is well written and requires only minor revisions prior to publication.

2. In the ethics statement in the manuscript and in the online submission form, please provide additional information about the patient records used in your retrospective study. Specifically, please ensure that you have discussed whether all data were fully anonymized before you accessed them and/or whether the IRB or ethics committee waived the requirement for informed consent. If patients provided informed written consent to have data from their medical records used in research, please include this information.

"MS and MH have received investigator initiated funding from Gilead Sciences, AbbVie and Bristol Myers Squibb for research unrelated to this work."

5. Please include your tables as part of your main manuscript and remove the individual files. Please note that supplementary tables (should remain/ be uploaded) as separate "supporting information" files.

Reviewers' comments:

Reviewer's Responses to Questions

**Comments to the Author**

1. Is the manuscript technically sound, and do the data support the conclusions?

Reviewer #1: Yes

Reviewer #2: Yes

2. Has the statistical analysis been performed appropriately and rigorously? 

Reviewer #1: Yes

Reviewer #2: Yes

3. Have the authors made all data underlying the findings in their manuscript fully available?

Reviewer #1: Yes

Reviewer #2: Yes

4. Is the manuscript presented in an intelligible fashion and written in standard English?

Reviewer #1: Yes

Reviewer #2: No

5. Review Comments to the Author

Reviewer #1: In the manuscript entitled “Phylogenetic clustering networks among heterosexual migrants with new HIV diagnoses post-migration in Australia”, the authors addressed the structure of phylogenetic network among migrants in Australia focusing on heterosexually infected individuals. The theme of the study is important and understudied. The study is clearly presented. The results are interesting and significant. The authors identified a differential clustering associated with cluster size. In small clusters – dyads, members were more likely to originate from the same country, while larger clusters were predominantly local and/or included members from different countries.

Minor critique:

- A brief comparison on the current status of HIV epidemic (e.g., rates of prevalence and/or incidence) between the study site, state Victoria, and other regions across Australia might provide an idea whether study results could be extrapolated to the whole country.

- Adding percentages might be helpful. For example, “Of these, 258 (30%) were in 92 phylogenetic clusters” gives the reader an idea about the proportion of clustered sequences. The authors provide some percentages, but there is some room for consistency.

- The authors are making a parallel with the Netherlands epidemic, which makes sense, but at the same time raises question whether other HIV transmission networks around the word could be used to better understand the underlying processes in the local HIV-1 epidemic in the Victoria state in Australia.

- The authors attempt to explain a large proportion of dyads by “heterosexually acquired infection among migrants mainly occurs within couples rather than larger sexual networks”. This link is questionable. Cluster size distribution demonstrates that dyads predominate in the vast majority of HIV transmission networks.

- The paragraph in Discussion before Limitations belongs rather to Introduction.

Reviewer #2: In their manuscript entitled ‘Phylogenetic clustering networks among heterosexual migrants with new HIV diagnoses post-migration in Australia, Sacks-Davis et al. used phylogenetic analysis of HIV-1 pol sequences of individuals infected with HIV-1 through heterosexual contacts and residing in the Australian state of Victoria. The major aim of the study was to look for indication for local transmission of the virus amongst migrants. The study is small but very well performed and the methodology used is appropriate. I particularly liked the approach to classify the evidence on probable place of HIV acquisition as ‘strong’ ‘medium’ or ‘weak’. This is an elegant way to minimize as much as possible a bias due to the difficulty to reliable assess the infection time and place.

The results of the study show no evidence of transmission networks in the migrant population. Transmissions between migrants were mostly limited to one-to-one transmissions. This observation confirms the findings of others.

Remarks:

One of the conclusions of the study is that the larger clusters that include members of foreign origin mixed with Australian individuals are probably driven by male-to-male sex. This too is an observation that has been reported before in other countries and in that regards it is a bit unfortunate that the authors have limited their investigations to individuals reporting heterosexual contacts as risk behavior. The study would greatly benefit from the additional inclusion of MSM.

In the Methods section, the study population is not well described. Numbers for the different populations should be mentioned here. This information is only found in the subsequent paragraph of Phylogenetic clustering. Also information is missing on the selection criteria for the non-migrant population selected?

The text may benefit from some re-editing. At least the following sentences need to be reformulated for better understanding: in the abstract: ‘Clusters of three or more …’; in the section on Statistical analysis: ‘on the basis of 859 sequences …..’; in the Discussion: ‘The eight clusters consisting….’.

Also, the authors almost systematically write a comma before ‘and’. There is no need to do this in case of enumerations, it inhibits a smooth reading. And I also would recommend to replace ‘heterosexual sex’ by ‘heterosexual contact’.

6. PLOS authors have the option to publish the peer review history of their article (what does this mean?). If published, this will include your full peer review and any attached files.

Reviewer #1: No

Reviewer #2: No

---

## [Author Response · Author response to Decision Letter 0]

13 Jul 2020

Reviewer 1 comments:

Comment 1: A brief comparison on the current status of HIV epidemic (e.g., rates of prevalence and/or incidence) between the study site, state Victoria, and other regions across Australia might provide an idea whether study results could be extrapolated to the whole country.

Author response: Additional detail has been added to the description of the HIV epidemic in Victoria in the Methods section (Study population subsection):

“Victoria is Australia’s second most populous state and has the second highest number of people diagnosed with HIV and new HIV diagnoses [25, 26].”

Comment 2: Adding percentages might be helpful. For example, “Of these, 258 (30%) were in 92 phylogenetic clusters” gives the reader an idea about the proportion of clustered sequences. The authors provide some percentages, but there is some room for consistency.

Author response: Additional percentages have been added to the Results section as requested.

Comment 3: The authors are making a parallel with the Netherlands epidemic, which makes sense, but at the same time raises question whether other HIV transmission networks around the word could be used to better understand the underlying processes in the local HIV-1 epidemic in the Victoria state in Australia.

Author response: Thank you to the reviewers for this point. To the best of our knowledge, the model from the Netherlands is the only study of HIV transmission networks in migrants globally. While there are many studies of HIV diagnoses and cascades of care in migrants, studies examining whether migrants acquired infection prior to or after migration, and studies of transmission networks among MSM and PWID, we are not aware of any other studies on HIV transmission networks in migrant communities. This has been clarified in the manuscript (Discussion section, page 17): 

“To the best of our knowledge, the only published study of HIV transmission networks in migrants is a mathematical modelling study of the heterosexual HIV epidemic in the Netherlands, which investigated the effects of sexual mixing patterns on the epidemic [24].”

If the reviewers are aware of other international studies that we have missed, we will add them to the manuscript.

Comment 4: The authors attempt to explain a large proportion of dyads by “heterosexually acquired infection among migrants mainly occurs within couples rather than larger sexual networks”. This link is questionable. Cluster size distribution demonstrates that dyads predominate in the vast majority of HIV transmission networks.

Author’s response: We agree with the reviewer that this sentence was misleading and have removed it from the Discussion. 

Comment 5: The paragraph in Discussion before Limitations belongs rather to Introduction.

Authors’ response: While most of this paragraph is a discussion of study findings, some of the details on previous research findings at the end of the paragraph were not directly linked to the study findings. We have removed some detail to make the connection between the latter part of the paragraph and study findings clearer.

Reviewer 2 comments:

Comment 1: One of the conclusions of the study is that the larger clusters that include members of foreign origin mixed with Australian individuals are probably driven by male-to-male sex. This too is an observation that has been reported before in other countries and in that regards it is a bit unfortunate that the authors have limited their investigations to individuals reporting heterosexual contacts as risk behavior. The study would greatly benefit from the additional inclusion of MSM.

Authors’ response: We agree with the reviewer that inclusion of MSM would facilitate investigation of homosexual sexual transmission of HIV among migrants. Unfortunately, we did not have access to that data when we completed this analysis. This is an area where we would like to conduct future research. 

Comment 2: In the Methods section, the study population is not well described. Numbers for the different populations should be mentioned here. This information is only found in the subsequent paragraph of Phylogenetic clustering. Also information is missing on the selection criteria for the non-migrant population selected?

Authors’ response: Consistent with the STROBE checklist for reporting of cross-sectional studies, we have included participant numbers in the Results section rather than the Methods section. These are illustrated in the flow chart (Figure 1) and described in paragraphs 1 and 2 of the Results. We have clarified the selection criteria for the non-migrant population in the Methods as requested (Study population section):

“…all notifications in non-migrants aged>18 years attributed to heterosexual sex and all notifications attributed to bisexual sex or injecting drug use (including those attributed to injecting drug use and MSM) in those >18 years irrespective of migrant status.”

Comment 3: The text may benefit from some re-editing. At least the following sentences need to be reformulated for better understanding: in the abstract: ‘Clusters of three or more …’; in the section on Statistical analysis: ‘on the basis of 859 sequences …..’; in the Discussion: ‘The eight clusters consisting….’.

Author response: Thank you for noting these difficult-to-read sentences. They have been amended as suggested. We have also re-edited the manuscript. 

Comment 4: Also, the authors almost systematically write a comma before ‘and’. There is no need to do this in case of enumerations, it inhibits a smooth reading. 

Author response: Most of these commas have been removed as requested.

Comment 5: I also would recommend to replace ‘heterosexual sex’ by ‘heterosexual contact’.

Author response: The HIV notification form refers to sexual contact. We have amended this and defined “heterosexual sex” as “heterosexual sexual contact” in the patient characteristics section of the Methods (page 6).

---

## [Decision Letter · Decision Letter 1]

28 Jul 2020

Phylogenetic clustering networks among heterosexual migrants with new HIV diagnoses post-migration in Australia

PONE-D-20-12180R1

Dear Dr. Sacks-Davis,

We’re pleased to inform you that your manuscript has been judged scientifically suitable for publication and will be formally accepted for publication once it meets all outstanding technical requirements.

Kind regards,

Jason Blackard, PhD

Academic Editor

PLOS ONE

Additional Editor Comments (optional):

None

Reviewers' comments:

Reviewer's Responses to Questions

**Comments to the Author**

1. If the authors have adequately addressed your comments raised in a previous round of review and you feel that this manuscript is now acceptable for publication, you may indicate that here to bypass the “Comments to the Author” section, enter your conflict of interest statement in the “Confidential to Editor” section, and submit your "Accept" recommendation.

Reviewer #1: All comments have been addressed

Reviewer #2: All comments have been addressed

2. Is the manuscript technically sound, and do the data support the conclusions?

Reviewer #1: Yes

Reviewer #2: Yes

3. Has the statistical analysis been performed appropriately and rigorously? 

Reviewer #1: Yes

Reviewer #2: N/A

4. Have the authors made all data underlying the findings in their manuscript fully available?

Reviewer #1: Yes

Reviewer #2: Yes

5. Is the manuscript presented in an intelligible fashion and written in standard English?

Reviewer #1: Yes

Reviewer #2: Yes

6. Review Comments to the Author

Reviewer #1: (No Response)

Reviewer #2: All remarks have been properly addressed. I have no further comments on the new version of the manuscript.

7. PLOS authors have the option to publish the peer review history of their article (what does this mean?). If published, this will include your full peer review and any attached files.

Reviewer #1: No

Reviewer #2: No

---

## [Editor Report · Acceptance letter]

17 Aug 2020

PONE-D-20-12180R1 

Phylogenetic clustering networks among heterosexual migrants with new HIV diagnoses post-migration in Australia 

Dear Dr. Sacks-Davis:

I'm pleased to inform you that your manuscript has been deemed suitable for publication in PLOS ONE. Congratulations! Your manuscript is now with our production department. 

Kind regards, 

on behalf of

Dr. Jason Blackard 

Academic Editor

PLOS ONE